# The association of weight status and weight perception with number of confidants in adolescents

Asuka Nishida[1], Jerome Clifford Foo[1,2], Shinji Shimodera[3], Atsushi Nishida[4], Yuji Okazaki[5], Fumiharu Togo[1], Tsukasa Sasaki[1] *

1 Department of Physical and Health Education, Graduate School of Education, The University of Tokyo, Tokyo, Japan, 2 Department of Genetic Epidemiology in Psychiatry, Central Institute of Mental Health, Medical Faculty Mannheim, University of Heidelberg, Mannheim, Germany, 3 Department of Neuropsychiatry, Kochi Medical School, Kochi University, Kochi, Japan, 4 Department of Psychiatry and Behavioral Science, Tokyo Metropolitan Institute of Medical Science, Tokyo, Japan, 5 Tokyo Metropolitan Matsuzawa Hospital, Tokyo, Japan

* psytokyo577@gmail.com

**Data Availability Statement:** All relevant data are within the manuscript and its Supporting Information files.

**Funding:** This study was supported by a grant from the Japan Society for the Promotion of

## Abstract

Weight status and self-weight perception are related to social relationship issues. Studies have suggested links between non-normal weight status or weight perception and youths having fewer confidants, but these relationships are unclear and remain to be studied. This preliminary cross-sectional study examined the effects of weight status and weight perception on the number of confidants in adolescents. Self-report data from 15,279 grade 7–12 students (54.2% boys) were analyzed. The number of confidants (0–3 or $\geq$ 4) was examined, according to five weight status categories (underweight, low-normal weight, mid-normal weight (reference), high-normal weight, overweight, with Body Mass Index corresponding to $\leq$ 18.5, $\leq$ 20.0, $\leq$ 22.5, $\leq$ 25.0 and > 25.0 in adults, respectively), and five weight perception categories (*too* thin, *a bit* thin, good (reference), *a bit* fat, *too* fat). Boys and girls who were overweight and those who perceived themselves to be *too* fat were significantly more likely to have few confidants. High-normal weight in girls and self-perception of being *a bit* fat in boys were also associated with having few confidants. In boys, underweight and self-perception of being *too* thin were additionally associated with having few confidants. Adolescents with non-normal weight status or weight perception may have fewer confidants and require more social support.

## Introduction

Adolescence is a sensitive developmental period in which youths acquire more sophistication in social interactions and experience changes in social expectations [1]. During adolescence, the prevalence of mental health problems increases sharply [2]. Among these problems are depressive disorders, self-harm, and anxiety disorders, which are some of the top ten leading causes of Disability-Adjusted Life Years [3] lost by adolescents [4]. One factor that protects adolescents from mental health problems is having access to confidants; people to whom these

Science (No. 15H03083) to TS. The funders had no
role in study design, data collection and analysis,
decision to publish, or preparation of the
manuscript.

**Competing interests:** The authors have declared
that no competing interests exist.

adolescents can talk about their problems. Adolescents who have few or no confidants are
more likely to have mental health problems such as higher levels of depressive symptoms [5–
7]. It has been reported that having 3 or fewer confidants is associated with the development of
mental health issues [8].

Overweight and obese adolescents often face social disadvantages in multiple domains of
living, including interpersonal relationships, as a result of weight stigmatization [9]. Weight
stigma portrays overweight and obese individuals with negative characteristics and beliefs,
such as lazy, stupid, less popular and lacking in friends [9]. Some evidence suggests that during
adolescence, being overweight/obese, which is highly stigmatized in our society, may compro-
mise their access to confidants. Overweight/obese adolescents may have smaller social network
sizes compared with non-overweight/obese peers. For example, being overweight/obese dur-
ing adolescence predicts social marginalization [10], including difficulty in making new
friends [11,12]. Overweight/obese adolescents are also less frequently nominated by their class-
mates as friends [11,13,14]. In addition, overweight/obese adolescents often have difficulty
perceiving their best friends as confidants [15] or have relationship difficulties with close
friends [12]. Overweight adolescents often report low quality of family interactions [16], such
as having difficulties in relationships with their parents [12], which may also contribute to
their lack of confiding relationships.

On the other hand, some studies have observed that being underweight can be related to
having less access to confidants. For example, underweight adolescents are more likely to have
problems in social behavior such as getting along with friends [17]. Underweight adolescents
also report a low frequency of family interactions, including a low frequency of meals eaten
together with family [18]. Furthermore, being underweight in boys has also been significantly
associated with having difficulty in discussing personal issues with parents [18]. A meta-analy-
sis which examined adults found that Body Mass Index (BMI) was significantly positively asso-
ciated with extraversion in males [19].

It is possible that weight perception of self (WP) may also affect confiding relationships in
adolescents, regardless of their weight status (WS). A systematic review observed that negative
body image (being both too thin and too fat) was often associated with lower levels of extraver-
sion in people of all ages [20]. Previous studies have also observed that adolescents who per-
ceive themselves as either being thin or fat report more social adaptation problems (e.g.
problems becoming accustomed to school life) [17,21] and impairment in social functioning
(e.g. trouble getting along with peers) [22]. In addition, a study observed that WP was discor-
dant with WS in a quarter of adolescents [23]. Furthermore, the effect of WP may not be in
accordance with the effect of the corresponding WS. For example, a perception of being either
thin or fat might not affect social relationships as negatively as actually being underweight or
overweight might, given that girls may be more likely to become friends with other girls who
have similar weight concerns [24].

According to these findings, WS and WP might be associated with the number of confiding
relationships in adolescents; these relationships, however, remain to be investigated. The pres-
ent study examines whether junior and senior high school students with non-normal WS and
WP have fewer confidants than their counterparts with normal WS and WP, using a self-
report questionnaire.

## Materials and methods

### Populations and procedures

A school-based cross-sectional survey targeting junior and senior high school students (Grades
7–12) was conducted from 2006 to 2009 in Japan. The principal investigators approached the

school principals of all public junior and senior high schools in Kochi prefecture (population: 780,000), and all public junior high schools in Tsu City (population: 290,000) in Mie prefecture. School principals were told that participation in the survey was voluntary. The principals then consulted with the teachers. The parents and guardians received a letter from the principal investigators that asked them to notify the school if they withheld consent for their child's participation in the present research. Among the 138 public junior and 36 public senior high schools, 45 junior high and 28 senior high schools participated in this study. Overall, 19,436 students were asked to participate in the self-reported survey. Of these, 798 were absent on the days of the survey and 388 declined to participate. Of the 18,250 students who agreed to participate, 2,234 students with missing data for height ($n = 839$) or weight ($n = 2,136$) were excluded. Furthermore, data from 175 students were excluded for not meeting a BMI value of 12–40 ($n = 125$) or not being within height threshold of 140–200 cm ($n = 50$) (i.e. invalid responses). Also, data from 416 students were excluded from the analysis because of incomplete answers to questions for WP ($n = 265$) or for the number of confidants ($n = 162$). In the end, data from 15,279 (83.7%; mean age = 15.3 ± 1.7; 54.2% boys) students were analyzed.

On the survey days, the teachers distributed to the students the survey questionnaire and an envelope (in which to seal the completed questionnaire before handing it back to the teachers). The students were told that participation in the study was anonymous and voluntary, and that answers would be kept confidential. Research staff collected the sealed questionnaires at each school. The study was conducted in accordance with Japan's Ethical Guidelines for Epidemiological Research. The data collection was approved by the ethics committees of the Tokyo Metropolitan Institute of Psychiatry (approval number: 20–9), the Mie University School of Medicine (approval number: 603), and Kochi Medical School at Kochi University (approval number: 20–57).

## Measurements

The questionnaire items and response options used in the present study are shown in S1 Table, including the measurement of independent and dependent variables and covariates in the original language and English. Details of the coding for each variable are described in the following sub-sections.

**Weight status and weight perception.**  Self-reported height and weight values were used to calculate BMI ($kg/m^2$). BMI values were initially split into four WS categories (underweight, normal weight, overweight, and obese), using BMI cut-off values that are equivalent to adult BMI values of 18.5, 25.0, and 30.0, based on age- and sex-specific cut-off values for under-weightedness [25] and overweightedness and obesity [26] in adolescents. The majority of subjects fell in the normal weight category. Thus, we further divided the normal weight category into low-normal weight, mid-normal weight, and high-normal weight using cut-off points which are comparable to the BMI values of 20.0 and 22.5 for adults [27–29]. Finally, the number of divisions for WS was reduced from 6 to 5 groups by merging the obesity group into the overweight group, since the number of obese subjects was small ($n = 224$ in boys; $n = 67$ in girls). In the end, the five WS categories were defined, with cut-off values adjusted for age and sex corresponding to the following equivalent BMIs in adults: Underweight: BMI < 18.5; Low-normal weight: $18.5 \leq BMI < 20.0$; Mid-normal weight: $20.0 \leq BMI < 22.5$; High-normal weight: $22.5 \leq BMI < 25.0$; Overweight: $BMI \geq 25.0$.

WP was measured using the following one-item question: "What do you think of your current body weight?" The students were given a five-point Likert scale with responses defined as follows: 1. "Too fat"; 2. "A bit fat"; 3. "Good"; 4. "A bit thin"; and 5. "Too thin". The students responded to the question by circling one of the given choices. In the analyses, each answer was considered as one category.

**Number of confidants.** The number of confidants was assessed with the following question, "How many people are there for you to confide in about your problems/concerns?", with five possible answers: 1. "None"; 2. "One"; 3. "Two"; 4. "Three"; 5. "Four or more". The students responded to the question by circling one of the given choices. Adolescents with 3 or fewer confidants were considered as having few confidants. A dichotomous variable was then created by combining answers of 1 to 4 into the "0–3 (Few) confidants" group, and putting answers of 5 in the "≥ 4 (More) confidants" group.

**Covariates.** Earlier studies found that non-normal WS significantly increases the risk of being bullied [30–32]. In addition, those who have past experience of being bullied often report low levels of support from friends [33], which may imply fewer confidants of the victims of bullying. Also, having few confidants has been reported to be significantly associated with depression and/or anxiety [5–7]. Our study entered these variables into the final analysis as confounders so that the risk of having few confidants in adolescents with non-normal WS/WP is examined regardless of the presence/absence of experience of being bullied and the current status of mental health. Experience of being bullied (yes or no) was assessed by asking the students, "Have you been bullied within the past year?" Depression/anxiety symptoms were measured using the Japanese version of the 12-item General Health Questionnaire (GHQ-12). The validity of GHQ-12 in adolescents has been previously confirmed [34]. In addition, the validity and reliability of the Japanese version of GHQ-12 has been established in people of all ages [35]. While each item of GHQ-12 was rated on a four-point Likert scale, the present study applied a bimodal scoring method (0-0-1-1); the Cronbach's alpha using this scoring method was 0.84 in the present sample. A score of 4 or higher in the possible score range of 0 to 12 is indicative of the risk of depression/anxiety in Japanese adults and adolescents [36]. These scores were entered as a dichotomous variable which were coded to 0 (scores from 0–3) and 1 (4–12) in the analyses.

## Statistical analysis

The associations of WS or WP with the number of confidants were tested using binary logistic regression. The number of confidants was specified as the dependent variable (Few: 0–3, More: ≥ 4), while either WS or WP was included as an independent variable. Analyses were adjusted for age, experience of being bullied (0 or 1) and GHQ-12 score (0 or 1). Mid-normal weight or perceiving oneself as good was chosen as the reference category in the analyses. All analyses were stratified by sex. In the adjusted analyses, 20 boys and 32 girls were excluded due to missing data about the experience of being bullied; a total of 8,088 boys and 7,139 girls were analyzed. Since age was a continuous variable, the present study conducted the Hosmer-Lemeshow test, a goodness of fit test for logistic regression, to assess how well the data fits the logistic regression models. It gave $p$-values of 0.43 and 0.75 for boys and girls respectively, indicating no evidence of poor fit. Testing the dichotomous variables gave $p$-value of 1. All statistical analyses were conducted using IBM SPSS Statistics version 25.0 for Microsoft Windows (IBM Corp., Armonk, NY).

## Results

### Descriptive statistics

The largest proportion of students (boys: 41.7%, girls: 40.1%) fell into the mid-normal weight group (Table 1). The prevalence of being underweight was 8.7% in boys and 12.7% in girls, while being overweight was 11.4% in boys and 6.6% in girls. WP for boys had a normal distribution, while the distribution of girls was skewed toward the perception of being fat. The distributions of WS, WP, and the number of confidants did not differ substantially across school

**Table 1. Weight status, weight perception, the number of confidants, experience of being bullied, and the General Health Questionnaire (GHQ)-12 scores in boys and girls.**

|  | Boys | Girls |
|---|---|---|
| N | 8,108 | 7,171 |
| **Age (years), mean ± SD** | 15.3 ± 1.7 | 15.3 ± 1.7 |
| **BMI (kg/m$^2$), mean ± SD** | 20.2 ± 3.2 | 19.9 ± 2.7 |
| **Weight status, n (%)** |  |  |
| Underweight | 716 (8.8) | 922 (12.9) |
| Low-normal weight | 1,610 (19.9) | 1,765 (24.6) |
| Mid-normal weight | 3,402 (42.0) | 2,882 (40.2) |
| High-normal weight | 1,456 (18.0) | 1,131 (15.8) |
| Overweight | 699 (8.6) | 404 (5.6) |
| Obese | 225 (2.8) | 67 (0.9) |
| **Weight perception, n (%)** |  |  |
| *Too* thin | 589 (7.3) | 87 (1.2) |
| *A bit* thin | 1,610 (19.9) | 328 (4.6) |
| Good | 3,291 (40.6) | 1,732 (24.2) |
| *A bit* fat | 1,910 (23.6) | 3,328 (46.4) |
| *Too* fat | 708 (8.7) | 1,696 (23.7) |
| **Number of confidant(s)[a], n (%)** |  |  |
| None | 1,844 (22.7) | 777 (10.8) |
| 1 | 744 (9.2) | 860 (12.0) |
| 2 | 1,257 (15.5) | 1,420 (19.8) |
| 3 | 822 (10.1) | 1,136 (15.8) |
| ≥ 4 | 3,441 (42.4) | 2,978 (41.5) |
| **Experience of being bullied (past 1 year), n (%)** |  |  |
| (+) | 621 (7.7) | 505 (7.0) |
| **GHQ-12 score, n (%)** |  |  |
| ≥ 4 | 2,699 (33.3) | 3,936 (54.9) |

n, number of subjects; SD, standard deviation.

[a] How many people are there for you to confide in about your concerns/problems?

level (Junior high / Senior high); in both boys and girls, the same category always showed the largest proportion within each variable. The distributions of WS, WP, and the number of confidants, stratified by school level are shown in S1–S4 Tables. As shown in Table 2, the distribution of WP in boys was relatively similar to the distribution for WS. However, a large proportion of girls who were underweight or normal weight reported perceiving themselves as fat. In both boys and girls, the highest prevalence of having 3 or fewer confidants was observed in overweight students.

## Binary logistic regression

The results for the binary logistic regression analyses are summarized in Table 3. Before adjustment for covariates, the odds ratios (ORs) were statistically significant for having few confidants in boys who were underweight ($p < .05$), low-normal weight ($p < .05$) or overweight ($p < .01$) and those who perceived themselves to be *too* thin ($p < .001$), *a bit* fat ($p < .001$), or *too* fat ($p < .001$). In girls, ORs were significant for those who were high-normal weight ($p < .001$) or overweight ($p < .001$) and those who perceived themselves to be *a bit* fat ($p < .05$) or *too* fat ($p < .001$). After adjusting for age, experience of being bullied and GHQ-12 scores, the

**Table 2. Weight perception and the number of confidants according to weight status, in boys and girls, *n* (%).**

| | Underweight | Low-normal weight | Mid-normal weight | High-normal weight | Overweight |
|---|---|---|---|---|---|
| | Boys (*N* = 8,108) | | | | |
| **Weight perception** | | | | | |
| *Too* thin | 239 (33.4) | 220 (13.7) | 115 (3.4) | 11 (0.8) | 4 (0.4) |
| *A bit* thin | 262 (36.6) | 586 (36.4) | 673 (19.8) | 78 (5.4) | 11 (1.2) |
| Good | 184 (25.7) | 648 (40.2) | 1,773 (52.1) | 590 (40.5) | 96 (10.4) |
| *A bit* fat | 24 (3.4) | 132 (8.2) | 759 (22.3) | 636 (43.7) | 359 (38.9) |
| *Too* fat | 7 (1.0) | 24 (1.5) | 82 (2.4) | 141 (9.7) | 454 (49.1) |
| **Number of confidants[a]** | | | | | |
| 0–3 | 432 (60.3) | 941 (58.4) | 1,886 (55.4) | 843 (57.9) | 565 (61.1) |
| ≥ 4 | 284 (39.7) | 669 (41.6) | 1,516 (44.6) | 613 (42.1) | 359 (38.9) |
| | Girls (*N* = 7,171) | | | | |
| **Weight perception** | | | | | |
| *Too* thin | 68 (7.4) | 14 (0.8) | 4 (0.1) | 1 (0.1) | 0 (0.0) |
| *A bit* thin | 200 (21.7) | 105 (5.9) | 22 (0.8) | 0 (0.0) | 1 (0.2) |
| Good | 408 (44.3) | 702 (39.8) | 557 (19.3) | 58 (5.1) | 7 (1.5) |
| *A bit* fat | 214 (23.2) | 794 (45.0) | 1,681 (58.3) | 547 (48.4) | 92 (19.5) |
| *Too* fat | 32 (3.5) | 150 (8.5) | 618 (21.4) | 525 (46.4) | 371 (78.8) |
| **Number of confidants[a]** | | | | | |
| 0–3 | 528 (57.3) | 1,008 (57.1) | 1,638 (56.8) | 709 (62.7) | 310 (65.8) |
| ≥ 4 | 394 (42.7) | 757 (42.9) | 1,244 (43.2) | 422 (37.3) | 161 (34.2) |

*n*, number of subjects.

[a] How many people are there for you to confide in about your concerns/problems?

odds ratios were statistically significant in boys who were underweight ($p < .05$) and overweight ($p < .05$) and those who perceived themselves as being *too* thin ($p < .05$), *a bit* fat ($p < .01$), and *too* fat ($p < .01$). In girls, being overweight ($p < .001$) and high-normal weight ($p < .01$) and perceiving oneself to be *too* fat ($p < .05$) showed significant odds ratios.

According to the sample calculation, the required sample size for the present binary logistic regression was shown to be 2,548 references versus 849 cases, when $\alpha = .05$ and $\beta = 0.1$, with the assumptions of the proportion of adolescents who have few confidants in the reference group = 0.4, OR = 1.3, and the ratio of the references versus cases = 3:1. When the ratio of the references versus cases is assumed = 1:1, the sample size required is calculated as 1,276 references versus 1,276 cases. The actual sample size, summarized in Table 1, seems to be close to the calculated sample size, except for the groups of "underweight" boys and "overweight (plus obese)" girls among WS categories (note that "obese" and "overweight" were merged into the "overweight" category in the analysis), and the groups of boys and girls perceiving themselves to be "*too* thin", girls perceiving themselves to be "*a bit* thin", and boys perceiving themselves to be "*too* fat" among WP categories. Among these groups, no significant associations between WS/WP and the number of confidants were observed in girls perceiving themselves to be "*a bit* thin" and "*too* thin".

## Discussion

This is the first study to examine the associations of WS and WP with the number of confidants in adolescents. Adolescents having non-normal WS or WP had fewer confidants. Both boys and girls who were overweight, when compared with mid-normal weight counterparts,

**Table 3. Odds ratios and 95% CIs for the effects of WS and WP on having few confidants when adjusted for age, experience of being bullied, and depression/anxiety symptoms.**

|  | Unadjusted | Adjusted |
|---|---|---|
| **Boys** | | |
| Underweight | **1.22 (1.04–1.44)** | **1.23 (1.04–1.45)** |
| Low-normal weight | **1.13 (1.00–1.27)** | *1.12 (0.99–1.27)* |
| High-normal weight | 1.10 (0.98–1.25) | 1.09 (0.96–1.24) |
| Overweight | **1.26 (1.09–1.47)** | **1.21 (1.04–1.41)** |
| *Too* thin | **1.37 (1.14–1.64)** | **1.21 (1.01–1.45)** |
| *A bit* thin | 1.03 (0.92–1.17) | 1.00 (0.90–1.13) |
| *A bit* fat | **1.33 (1.19–1.49)** | **1.21 (1.07–1.36)** |
| *Too* fat | **1.54 (1.30–1.83)** | **1.31 (1.10–1.56)** |
| **Girls** | | |
| Underweight | 1.01 (0.87–1.17) | 1.01 (0.86–1.17) |
| Low-normal weight | 1.01 (0.90–1.14) | 1.00 (0.89–1.13) |
| High-normal weight | **1.28 (1.11–1.47)** | **1.26 (1.09–1.46)** |
| Overweight | **1.45 (1.18–1.78)** | **1.41 (1.15–1.74)** |
| *Too* thin | 1.43 (0.92–2.24) | 1.31 (0.83–2.06) |
| *A bit* thin | 1.04 (0.82–1.33) | 0.99 (0.78–1.27) |
| *A bit* fat | **1.15 (1.02–1.29)** | 1.05 (0.93–1.18) |
| *Too* fat | **1.42 (1.24–1.63)** | **1.19 (1.03–1.37)** |

CI, confidence interval; WS, weight status (reference: mid-normal weight); WP, weight perception (reference: perceiving one's weight to be good). Significant associations are indicated in bold ($p < .05$) and italic ($p < .10$) letters. Covariates include age, experience of being bullied, and GHQ-12 score (0/1).

or perceived themselves as being *too* fat, when compared with those who perceived their weight to be good, were statistically significantly more likely to have few confidants. Girls who were high-normal weight were also more likely to have few confidants, as were boys who were underweight or perceived themselves to be *too* thin or *a bit* fat. Most associations remained significant after controlling for depressive and anxiety symptoms, suggesting that adolescents with non-normal WS or WP, even when they are currently without the symptoms, may be more likely to have few confidants compared with their reference counterparts.

Overweight adolescents being more likely to have few confidants is in line with studies in the literature. A previous study reported that overweight adolescents often do not have close friends or have difficulties in family relationships [12]. Another study found that overweight adolescents often do not perceive their best friends as confidants [15]. One study reported that overweight girls are less likely than normal weight counterparts to perceive their family members as confidants [16]. The reasons for overweight adolescents having fewer confidants may be partly explained by psychosocial factors including stigmatization. Overweight adolescents may often experience social rejection since individuals, including adolescents [37], often prefer thinner people [38]. High stigma with respect to body weight may make overweight adolescents opt out of a broader peer network [39], leading to less time spent with friends after school [12,15,17,22]. Overweight adolescents are also more likely to make friends with overweight peers [40] or peers with similar levels of bulimic behavior [24], which can limit their sources of friendships. Girls who had high-normal weight in the present study were also more likely to have few confidants. High-normal weight girls, as well as overweight girls, may experience social rejection by peers since teenage girls often characterize their ideal weight as low-normal weight [41].

Adolescents who perceived themselves to be *too* fat were more likely to have few confidants. This may be in line with previous studies finding that perceived-overweightedness was associated with social relationship difficulties [17,21] and withdrawnness [17] in adolescents and that perceived-overweightedness could be predicted by receiving negative comments from peers [42]. Similarly, boys with the perception that they were *a bit* fat were also more likely to have few confidants, but this was not observed in girls. A previous study reported that girls are more likely to become friends with other girls who have similar weight/shape concerns [24]. In the present study, more than half of the girls reported perceiving themselves to be *a bit* fat. In boys, the proportion was half of that in girls. This may suggest that the perception of being *a bit* fat may actually be "normal" in girls, mitigating any negative effects on the number of confidants, but not in boys.

The present study also showed that boys who were underweight or perceived themselves to be underweight were more likely to have few confidants. This may be in line with previous findings that males who are underweight are more likely to be introverted [19], and that boys who are extremely thin and who perceive themselves to be underweight have difficulty confiding in parents about personal issues [18]. While girls typically wish to be thinner, in contrast, boys often want to be larger [43]. In boys, it has been shown that lower WS predicted a stronger increase in muscularity concerns, referred to as muscular-ideal internalization, compared with heavier weight counterparts [44]. This desire for muscularity might have confounded the present observation in boys. Indeed, muscularity concerns significantly increase during adolescence [37,44] and muscle-gaining behaviors are shown to be more salient in popular boys [37]. Our findings raise the importance of recognizing not only overweightedness and obesity but also underweightedness as an issue regarding social relationships among adolescents.

WS and WP were significantly associated with having few confidants, after controlling for current depressive and anxiety symptoms. Having few confidants has been associated with subsequent development of mental health problems, including depressive disorder [45] and attempted self-harm [46]. The present results, taken together with these observations, suggest that adolescents with non-normal WS or WP might have increased risk of developing mental health problems associated with having few confidants, even if they do not currently have mental health problems. This seems to be in accordance with some previous studies, which find adverse effects of WS and WP on later developing mental health problems [47–52].

## Limitations

Our study has several limitations. First, this study used cross-sectional data, and the results might not explain causality. Second, BMI values were calculated using self-report height and weight, which may have affected estimations of BMI values in some groups; a recent meta-analysis observed that, at least for the screening of overweight and obesity in adolescents, self-report data is a valid alternative to directly measured height and weight [53]. Third, outcomes were measured using a single question. Single questions may be inadequate to accurately assess participants' characteristics. For example, adolescents might have different reasons for their WP, including muscularity concerns and shape concerns about specific body parts. The present study also did not elaborate on which type of confiding relationships, such as family, friends, or teachers, underlay the number of confidants. Future studies on WS and WP should also take into account whether the number of friends or social network size may affect the number of confidants a person has, since a lower number of confidants may be a side effect of loss of friends due to experiencing social rejection, for example through bullying and weight-related marginalization [31,32]. Fourth, inadequate sample size for girls perceiving themselves to be *too* thin or *a bit* thin might have affected their associations with the number of confidants due to the lack of statistical power.

## Conclusions

The present study provides a novel perspective for understanding adolescents' social and psychological problems in the light of body weight status and weight perception. As adolescents with non-normal WS and WP appear likely to have few confidants, more effort should be made to recognize and understand issues around social relationships in these adolescents and to give them more support in dealing with their challenges. Not only peers, but also family members and teachers should be made aware of the present findings and encouraged to look out for these adolescents.

## Supporting information

**S1 Table. Measurement of independent and dependent variables and covariates in the original language and English.**
(PDF)

**S2 Table. The distribution of weight status in boys and girls, stratified by school level, *n* (%).** *n*, number of subjects.
(PDF)

**S3 Table. The distribution of weight perception in boys and girls, stratified by school level, *n* (%).** *n*, number of subjects.
(PDF)

**S4 Table. The distribution of the number of confidants in boys and girls, stratified by school level, *n* (%).** *n*, number of subjects.
(PDF)

## Acknowledgments

The authors thank Dr. Satoshi Usami at Center for Research and Development on Transition from Secondary to Higher Education, the University of Tokyo, for providing us direct technical help and expertise on statistics. The authors are grateful to all the junior and senior high school students who put time and effort into completing the survey, and to the school teachers for their great support with data collection.

## Author Contributions

**Conceptualization:** Asuka Nishida.

**Data curation:** Shinji Shimodera, Atsushi Nishida, Yuji Okazaki.

**Formal analysis:** Asuka Nishida, Fumiharu Togo, Tsukasa Sasaki.

**Funding acquisition:** Tsukasa Sasaki.

**Methodology:** Fumiharu Togo.

**Supervision:** Tsukasa Sasaki.

**Writing – original draft:** Asuka Nishida, Tsukasa Sasaki.

**Writing – review & editing:** Asuka Nishida, Jerome Clifford Foo, Fumiharu Togo, Tsukasa Sasaki.

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
