## [Decision Letter · Decision Letter 0]

29 Aug 2019

PONE-D-19-18018

The association of weight status and weight perception with number of confidants in adolescents

PLOS ONE

Dear Prof. Sasaki,

Thank you for submitting your manuscript to PLOS ONE. After careful consideration, we feel that it has merit but does not fully meet PLOS ONE’s publication criteria as it currently stands. Therefore, we invite you to submit a revised version of the manuscript that addresses the points raised during the review process.

The reviewers addressed several major and minor concerns about your manuscript. Please revise your manuscript carefully.

We would appreciate receiving your revised manuscript by Oct 13 2019 11:59PM. To enhance the reproducibility of your results, we recommend that if applicable you deposit your laboratory protocols in protocols.io, where a protocol can be assigned its own identifier (DOI) such that it can be cited independently in the future. For instructions see: http://journals.plos.org/plosone/s/submission-guidelines#loc-laboratory-protocols

We look forward to receiving your revised manuscript.

Kind regards,

Kenji Hashimoto, PhD

Academic Editor

PLOS ONE

Journal Requirements:

Reviewers' comments:

Reviewer's Responses to Questions

**Comments to the Author**

1. Is the manuscript technically sound, and do the data support the conclusions?

Reviewer #1: No

Reviewer #2: Yes

Reviewer #3: Yes

2. Has the statistical analysis been performed appropriately and rigorously? 

Reviewer #1: No

Reviewer #2: Yes

Reviewer #3: Yes

3. Have the authors made all data underlying the findings in their manuscript fully available?

Reviewer #1: Yes

Reviewer #2: Yes

Reviewer #3: No

4. Is the manuscript presented in an intelligible fashion and written in standard English?

Reviewer #1: Yes

Reviewer #2: Yes

Reviewer #3: Yes

5. Review Comments to the Author

Reviewer #1: The manuscript entitled “The association of weight status and weight perception with number of confidants in adolescents” presents interesting data, however manuscript can not be published because inadequate methodology has been applied.

Authors presented BMI cut-off by WHO, however this values are applicable only for adult population. For children growth reference charts must be applied – e.g. WHO recommendations for children under 18 years old. If in Japan national charts exist, it is an even better situation because of a better fitting for the Japanese adolescent population. Therefore the national growth reference charts should be applied.

Moreover, Authors should present the number of ethics committee's approval.

Authors also should complete the information about the statistical tests that have been applied. How did Authors verify the normality of data distribution? What tests did they used?

In my opinion, anthropometric measurements should be made. Especially, taking into account that another authors indicate that more than half middle- and high-school students, did not correctly perceive their own body weight (Yan et al. 2018. Body Weight Misperception and Its Association with Unhealthy Eating Behaviors among Adolescents in China. IJERPH.). In such situation, the own perception of body mass can not be perceived as a valid measurement.

Reviewer #2: 1. Please include explanation on why WS and WP were both used in the study in the Introduction Section.

2. Please state the sampling method used in the Methodology Section.

3. Please justify the division of weight category into low-normal weight, mid-normal weight, and high-normal weight (Line 132).

4. Please state the Operational Definition of 'WS, WP and Confidants' in the Methodology.

5. Since this study involved adolescents grade 7-12 (junior and high schools), was there any difference in terms of WS, WP and confidants numbers between age groups or school levels? Please elaborate more on age factor in the discussion.

Reviewer #3: Thank you for the opportunity to review this interesting study on the association between non-normal weight status or weight perception and the number of confidants in adolescents. The manuscript addresses a relevant and important topic for the study of the health of adolescents.

Abstract:

I suggest including design of study in abstract.

Introduction:

1. The introduction is adequate, however I suggest excluding this sentence “Analyses of the associations are stratified by sex and further adjusted for age, experience of being bullied, and depressive and anxiety symptoms.”

Methods

1. Was there sample calculation? Please make it clear.

2. Why the authors decided to use Cole et al (2000 and 2007) as the reference of BMI cut-off? These BMI cut-off values are adequate to evaluate anthropometric status in children and adolescents, but have as limitations to use the same cut-offs for men and women.

3. Were the tools validated for adolescents? Could you indicate the alpha data? Please complete this.

Results:

1. I suggest avoiding the use of single-sentence paragraphs, for instance “The largest proportion of students (boys: 41.7%, girls: 40.1%) fell into the mid-normal weight group (Table 1).”

2. I suggest writing this sentence “The prevalence of being underweight 177 was 8.7% in boys and 12.7% in girls, while being overweight was 14.2% in boys and 7.5% in girls. WP for boys had a normal distribution, while the distribution of girls was skewed toward the perception of being fat.” before table 1.

Discussion:

1. In the first paragraph include for boys the association found between to be a bit fat and to have few confidants among adolescents.

2. I suggest improving the discussion for the result presented on pages 16 and 17, lines 231 to 236, about the association found between feeling a bit fat and to have few confidants among boys.

Conclusion:

Conclusions are adequate.

6. PLOS authors have the option to publish the peer review history of their article (what does this mean?). If published, this will include your full peer review and any attached files.

Reviewer #1: No

Reviewer #2: No

Reviewer #3: No

---

## [Author Response · Author response to Decision Letter 0]

16 Oct 2019

Thank you all very much for your constructive comments. We have carefully considered all the points addressed and the feedback and have accordingly made substantial changes to the manuscript. Responses to each point raised by the Academic Editor and the Reviewers are addressed in the attachment file labeled 'Response to Reviewers'.

Please let us know if any additional information or changes are needed. We look forward to working with you to bring our manuscript to publication.

---

## [Decision Letter · Decision Letter 1]

30 Oct 2019

PONE-D-19-18018R1

The association of weight status and weight perception with number of confidants in adolescents

PLOS ONE

Dear Prof. Sasaki,

Thank you for submitting your manuscript to PLOS ONE. After careful consideration, we feel that it has merit but does not fully meet PLOS ONE’s publication criteria as it currently stands. Therefore, we invite you to submit a revised version of the manuscript that addresses the points raised during the review process.

Two reviewers addressed several minor concerns about your revised manuscript. Please revise your manuscript again.

We would appreciate receiving your revised manuscript by Dec 14 2019 11:59PM. To enhance the reproducibility of your results, we recommend that if applicable you deposit your laboratory protocols in protocols.io, where a protocol can be assigned its own identifier (DOI) such that it can be cited independently in the future. For instructions see: http://journals.plos.org/plosone/s/submission-guidelines#loc-laboratory-protocols

We look forward to receiving your revised manuscript.

Kind regards,

Kenji Hashimoto, PhD

Academic Editor

PLOS ONE

Reviewers' comments:

Reviewer's Responses to Questions

**Comments to the Author**

1. If the authors have adequately addressed your comments raised in a previous round of review and you feel that this manuscript is now acceptable for publication, you may indicate that here to bypass the “Comments to the Author” section, enter your conflict of interest statement in the “Confidential to Editor” section, and submit your "Accept" recommendation.

Reviewer #1: (No Response)

Reviewer #2: All comments have been addressed

Reviewer #3: All comments have been addressed

2. Is the manuscript technically sound, and do the data support the conclusions?

Reviewer #1: Yes

Reviewer #2: Yes

Reviewer #3: Yes

3. Has the statistical analysis been performed appropriately and rigorously? 

Reviewer #1: No

Reviewer #2: Yes

Reviewer #3: Yes

4. Have the authors made all data underlying the findings in their manuscript fully available?

Reviewer #1: Yes

Reviewer #2: Yes

Reviewer #3: Yes

5. Is the manuscript presented in an intelligible fashion and written in standard English?

Reviewer #1: Yes

Reviewer #2: Yes

Reviewer #3: Yes

6. Review Comments to the Author

Reviewer #1: Authors need to verify the data distribution and must use the appropriate statistical tests. Using other tests can change the Results, and thus the Discussion and Conclusion. How did Authors verify the normal distribution of data? What tests they used? Authors should complete this information.

Reviewer #2: Thank you for responding accordingly to each comment. All of them have been addressed well. Congratulations on your rigorous analysis.

Reviewer #3: Thanks for answering all the questions, however, one question in the introduction section and two in the method still require minor adjustments as the following.

Introduction:

Although the authors agree to remove the sentence “Analyses of the associations are stratified by sex and further adjusted for age, experience of being bullied, and depressive and anxiety symptoms.” of the Introduction and report deleting it, it still remains in the manuscript in the page 5, lines 95 and 96.

Methods

1. I suggest removing the sampling process of the Results Section and including it in the Methodology Section

2. I apologize because I didn’t give the correct information. Was the General Health Questionnaire validated for adolescents? Could you indicate the alpha data? Please complete this.

Regards

7. PLOS authors have the option to publish the peer review history of their article (what does this mean?). If published, this will include your full peer review and any attached files.

Reviewer #1: No

Reviewer #2: No

Reviewer #3: No

---

## [Author Response · Author response to Decision Letter 1]

14 Nov 2019

Thank you all very much for your constructive comments. We have carefully considered all the points addressed and the feedback and have made changes to the manuscript accordingly. Responses to each point raised by the Reviewers are addressed in the attachment file labeled 'Response to Reviewers'.

Please let us know if any additional information or changes are needed. We look forward to working with you to bring our manuscript to publication.

---

## [Editor Report · Decision Letter 2]

15 Nov 2019

The association of weight status and weight perception with number of confidants in adolescents

PONE-D-19-18018R2

Dear Dr. Sasaki,

We are pleased to inform you that your manuscript has been judged scientifically suitable for publication and will be formally accepted for publication once it complies with all outstanding technical requirements.

With kind regards,

Kenji Hashimoto, PhD

Section Editor

PLOS ONE
---

## [Editor Report · Acceptance letter]

20 Nov 2019

PONE-D-19-18018R2 

The association of weight status and weight perception with number of confidants in adolescents 

Dear Dr. Sasaki:

I am pleased to inform you that your manuscript has been deemed suitable for publication in PLOS ONE. Congratulations! Your manuscript is now with our production department. 

With kind regards,

on behalf of

Prof. Kenji Hashimoto 

Section Editor

PLOS ONE